# Selection and Validation of Reference Genes in Different Tissues of Okra (*Abelmoschus esculentus* L.) under Different Abiotic Stresses

**DOI:** 10.3390/genes14030603

**Published:** 2023-02-27

**Authors:** Zhipeng Zhu, Jianxiang Yu, Xinhui Tang, Aisheng Xiong, Miao Sun

**Affiliations:** 1College of Marine and Biological Engineering, Yancheng Teachers University, Yancheng 224002, China; 2State Key Laboratory of Crop Genetics and Germplasm Enhancement, College of Horticulture, Nanjing Agricultural University, Nanjing 210095, China

**Keywords:** *Abelmoschus esculentus* L., reference gene, RT-qPCR, abiotic stress

## Abstract

Okra (*Abelmoschus esculentus* L.) is a particular vegetable with both edible and medicinal values. However, the expression pattern of the okra reference genes in response to abiotic stress has not been explored. In the present study, 18 potential reference genes were selected from okra in various tissues and abiotic stress conditions, and their expression levels were detected by Real-Time quantitative PCR (RT-qPCR). Their expression stabilities were calculated by four algorithms (geNorm, NormFinder, BestKeeper, and RefFinder). Under cold stress, the most stable genes included *GAPC1* and *CYP* (leaf), *CYP* and *ACT7* (root), *HIS6* and *GAPC1* (stem), and *HIS6* and *60s* (different tissues). Under salt stress, *EF-1α* and *UBQ* (leaf), *EF-1α* and *UBQ* (root), *TUA4* and *Eif* (stem), and *HIS6* and *Eif* (different tissues) were the most stable genes. Under drought stress, *UBQ* and *Eif* in the leaf, *HIS6* and *Eif* in the root, *TUA4* and *HIS6* in the stem, and *UBQ* and *Eif* in different tissues were most stably expressed in okra. In addition, complete sequencing results by RefFinder showed that *HIS6* and *ACT7* in the leaf, *HIS6* and *Eif* in the root, *UBC5B* and *60s* in the stem, and *HIS6* and *Eif* in different tissues, were most the suitable reference genes for okra. Furthermore, *AeMYB1R1* transcription factor was used to verify the reliability of RT-qPCR values. In summary, this study was carried out to demonstrate the potential reference genes of okra under abiotic stress, aiming to provide a molecular basis for functional gene analysis and regulatory mechanism research of okra.

## 1. Introduction

Okra (*Abelmoschus esculentus* L.) belongs to the *Malvaceae* family and is widely cultivated worldwide [1]. Okra has been reported to have excellent edible [2] and nutritional [3] values. Moreover, okra has an essential medicinal value [4]. It can be used for anti-inflammatory [5], antioxidant [6], treatment of kidney diseases [7], and liver protection [8]. However, as the environment deteriorates, various abiotic stresses have serious effects on okra production [9].

Drought stress, cold stress, and salt stress are serious and sweeping abiotic stress factors on plants, which directly lead to crop decline worldwide [10,11]. Drought stress can cause a series of morphological, physiological, and biochemical changes in plants such as loss of leaf water, inhibition of plant growth, destruction of cell membrane structure, and electrolyte extravasation [12]. The damage caused by cold stress to plants is mainly manifested in changes in membrane lipid phase, increase in membrane permeability, increase in reactive oxygen species (ROS) and free radicals, decrease in chlorophyll content, and the accumulation of secondary metabolites [13]. Salt stress will reduce soil water potential and severely hinder plant growth and development by causing osmotic stress and destruction of ion balance, leading to plant malnutrition and membrane lipid peroxidation [14]. Nowadays, benefiting from genetic technology, including CRISPR/Cas, RNAi, and VIGS, the expression pattern of plant functional genes under abiotic stress is fully understood [15]. RT-qPCR is a standard technology for determining gene expression levels under abiotic stress, characterized by strong specificity, high sensitivity, accurate quantification, good repeatability, rapid response, and high throughput. However, RNA and cDNA quality, PCR amplification efficiency, and primer specificity may affect the accuracy of RT-qPCR results [16]. Therefore, suitable reference genes should be screened, and the expression levels of functional genes could be accurately analyzed by RT-qPCR [17]. Plant reference genes are stable in all samples and conditions, such as *ACT*, *TUB*, and *EF-1α* [18]. To date, reference genes of maize [19], sorghum [20], tomato [21], and other plants have been recorded in the ICG knowledge base (http://icg.big.ac.cn, accessed on 10 May 2022). However, there are few reports on okra reference genes [22]. Therefore, it is vital to explore and identify the potential reference genes of okra under various abiotic stresses.

Under different species and treatments, commonly used reference genes show significant expression instability [23]. Therefore, to obtain accurate RT-qPCR data, appropriate reference genes should be screened under specific conditions [24]. When selecting and verifying candidate reference genes, algorithms such as geNorm [25], NormFinder [26], BestKeeper [27], and RefFinder [28] are widely used in further analysis of RT-qPCR data [29,30]. To date, only a small number of studies have verified several reference genes for transcription normalization [31]. In this study, 18 potential reference genes were screened out from our published okra transcriptomic database [1] (NCBI accession number: SRR13983395). Gene expression analysis was performed on different tissues under various abiotic stresses by RT-qPCR to screen for the most suitable reference genes. Experimental groups in this study included three types of tissue (root, stem, and leaf) and three abiotic stress conditions (drought, salt, and cold stress). The 18 genes selected are common reference genes in plants, including *Toona Ciliata* [23], *Glycine max* [32], and *Gentiana Macrophylla* [33]. The genes in this study were actin-7 (*ACT7*), histone acetyltransferase MCC1 (*HIS1*), histone deacetylase 6 (*HIS6*), ubiquitin-conjugating enzyme E2 5B (*UBC5B*), ubiquitin-conjugating enzyme E2–17 kDa (*UBC17*), tubulin α-3 chain (*TUB-α*), S-adenosylmethionine decarboxylase proenzyme (*SAMDC*), membrane-anchored ubiquitin-fold protein 1 (*MUB1*), cyclophilin (*CYP*), polyubiquitin (*UBQ*), eukaryotic translation initiation factor (*Eif*), elongation factor 1 α (*EF-1α*), ADP-ribosylation factor A1E (*ARFA1E*), Glyceraldehyde-3-phosphate dehydrogenase C subunit 1 (*GAPC1*), Tubulin α-4 chain (*TUA4*), actin-related protein 2 (*ARP*), glyceraldehyde-3-phosphate dehydrogenase (*GAPDH*), and 60s ribosomal protein l 36 (*60s*). In addition, *AeMYB1R1* transcription factor was used to verify the expression level of 18 selected genes. MYB, one of the most prominent transcription factor families in plants, regulates a variety of biological processes, including specific development, metabolism, response to abiotic stresses [34,35] in *Arabidopsis thaliana* [36], *Zea mays* [37], and *G. max* [38]. *MYB1R1* is involved in gene regulation under stress in plants [39,40]. To validate the best selected reference genes, *AeMYB1R1* expression levels under different conditions were investigated using the most stable and least stable reference genes or combinations. This study aimed to provide suitable reference genes for okra under different abiotic stresses and lay a theoretical foundation for improving okra yield and quality.

## 2. Materials and Methods

### 2.1. Plant Materials, Treatments, and Sample Collecting of Okra

The plant molecular biology laboratory of Yancheng Teachers University (33°39′ N, 120°16′ E) preserved okra seeds (*A. esculentus* L.) (cv ‘Hokkaido No.1’). After germination, okra seeds were transferred to nutrient soil for further cultivation. The growth condition was maintained at 22 °C/20 °C (day/night), 16 h/8 h (day/night) supplemental illumination with 300 μmol·(m^2^·s)^−1^ light intensity. One-month-old seedlings were treated with cold (−7 °C treatment), drought (20% PEG6000 solution treatment), and salt (200 mM NaCl solution treatment) stress. Experimental materials were processed three times and sampled at 0 h (10 am), 6 h (4 pm), 12 h (10 pm), and 24 h (10 am). The leaf, root, and stem samples of okra were separated, washed, and stored in a −80 °C ultralow temperature freezer for further analysis (Table 1). In all cases, three or more independent biological replicates were obtained from each sampling point.

### 2.2. Total RNA Extraction and cDNA Synthesis of Okra Samples

Total RNA was extracted from okra samples via Tiangen EasyPure^®^ RNA Kit (Catalog Number: ER101-01, purchased from Beijing Transgen Biotechnology Co., Ltd., Beijing, China). RNA quality and RNA quantity were measured in the Thermo Fisher Scientific Spectrophotometer (Catalog Number: 912A0959, purchased from Thermofisher Scientific (China) Co., Ltd., Shanghai, China). RNA integrity was assessed by electrophoresis on 1.0% agarose gel. The RNA concentration was measured using a Thermo Scientific™ NanoDrop Lite (Catalog Number: ND-LITE-PR, purchased from Thermofisher Scientific (China) Co., Ltd., Shanghai, China).

The cDNA of okra samples was synthesized according to the product manual of the PrimeScript™ RT Master Mix (Catalog Number: RR036Q, purchased from Beijing Takara Biomedical Technology Co., Ltd., Beijing, China).

### 2.3. The Screening of Potential Reference Genes and Primer Design for RT-qPCR

Based on previous research and our published *A. esculentus* transcriptomic database (NCBI accession number: SRR13983395), 18 potential reference genes were screened, including *ACT7*, *HIS1*, *HIS6*, *UBC5B*, *UBC17*, *TUB-α*, *SAMDC*, *MUB1*, *CYP*, *UBQ*, *Eif*, *EF-1α*, *ARFA1E*, *GAPC1*, *TUA4*, *ARP*, *GAPDH*, and *60s*. The nucleotide sequences of these 18 genes were obtained from the transcriptomic database. The primers for RT-qPCR were designed by NCBI Primer-Blast tool and DAMAN 6.0 software (Table 2) and synthesized by Anhui General Biological Systems Co., Ltd. (Chuzhou, China).

### 2.4. PCR and RT-qPCR Analysis

The PCR reaction was performed with TaKaRa PCR Amplification Kit (Catalog Number: R011, purchased from Beijing Takara Biomedical Technology Co., Ltd., Beijing, China) to test all primer pairs for specificity. The amplified products were analyzed by 1.0% agarose gel electrophoresis and visualized in a Gel Doc XR+ Gel Documentation System (Catalog Number: 1708195, purchased from Bio-Rad Laboratories, Hercules, CA, USA).

Three biological and three technical replicates were run for each RT-qPCR run. RT-qPCR was performed by SYBR^®^ Premix Ex Taq^TM^ II (Catalog Number: RR820Q, purchased from Beijing TaKaRa Biotechnology Co., Ltd., Beijing, China) and performed on a Thermal Cycler Dice^TM^ Real-Time System *Lite* (Code No. TP700/TP760, purchased from TaKaRa Biotechnology (Dalian) Co., Ltd., Dalian, China). The PCR reaction procedure was as follows: 95.0 °C for 2.0 min and 40 cycles of 95.0 °C for 15.0 s, 60.0 °C for 15.0 s. The melting curve was drawn to confirm the specific RT-qPCR reaction product of three technical replicates and one template-free control. For each condition, a standard curve was generated using 3-fold serial dilutions of cDNA in triplicate. the amplification efficiency of each primer was calculated according to the formula
E = (10^(−1/slope)^ − 1) × 100.

In addition, a standard curve was established and used to calculate PCR efficiency (E (%)) and correlation coefficient (R^2^) [41].

### 2.5. Ranking of Reference Gene Expression Stability Based on Four Algorithms

To evaluate and rank the expression stability of the 18 genes, three commonly used algorithms (geNorm, NormFinder, BestKeeper) were applied in multiple experimental groups. Using geNorm, an appropriate number of reference genes can be estimated by analyzing the pairing difference (Vn/n+1) of normalization factors. Vn/n+1 ≤ 0.15 means that the correct number of reference genes is n, and if the value is more than 0.15, another reference gene should be added to normalization factors until the value reaches 0.15. NormFinder determines the ranking of reference genes by calculating the stability of gene expression. The relative expressions of 2^−ΔCt^ of each gene in each sample were used as the raw data to generate the stable value of gene expression, and then the most stable gene was selected according to the most stable value. BestKeeper determines gene expression stability by counting SD and CV values. After that, RefFinder is used to summarize the ranking and obtain the complete order of these 18 genes [28].

### 2.6. Accuracy Assessment of Reference Genes by the Transcription Factor AeMYB1R1

To evaluate the reliability of the stability of the expression of the reference gene with *AeMYB1R1* transcription factor as the target gene, under cold, salt, and drought stress, and in different tissues, the relative expression levels of the top two and bottom two reference genes in the RefFinder stability ranking table (Appendix A) were detected. In addition, three technical replicates were carried out. At the same time, relative gene expression levels were calculated and analyzed using the method described by Schmittgen and Livak [42].

## 3. Results

### 3.1. Amplification Specificity and Efficiency of Reference Genes in Okra

All reference genes had a single band in the gel, and RT-qPCR melting curves showed that each amplified candidate reference gene had a single peak (Appendix A). These results indicated that the primers had a high amplification specificity. The linear correlation coefficients of the melting curves of 18 genes were all greater than 0.99 (R^2^ > 0.99), indicating that the RT-qPCR data had a high degree of fitting. Amplification efficiency data of 18 genes showed that the lowest value was 90.82% (*UBQ*), and the highest value was 102.84% (*TUA4*), meeting the analysis requirements of plant reference genes (Table 2).

### 3.2. The Expression Profile Analysis of Reference Genes in Okra

The expression profiles of 18 genes were measured under the following conditions: different okra tissues (leaf, root, and stem), cold, salt, and drought stress. The data showed that the expression profiles of 18 genes varied significantly among different samples. Among them, the highest and lowest CT values were 33.89 and 16.64 (Figure 1). The most abundant gene was *EF-1α*, whose maximum, minimum, and median CT values were 32.61, 19.07, and 26.36, respectively. The least abundantly expressed gene was *UBQ*, with maximum, minimum, and median CT values of 22.16, 20.18, and 21.17, respectively.

Compared with other reference genes, the CT value ranges of *UBQ*, *HIS6*, and *Eif* were relatively narrow, indicating that their expression was more stable in different samples. In addition, the expression of 18 genes showed significant variability.

### 3.3. geNorm Analysis of Reference Genes in Okra

The expression stabilities of 18 genes from okra were analyzed by geNorm. In this algorithm, the M value represents the basis of the evaluation of gene expression stability (M < 1.0) [43]. Gene expression levels with lower M values are more stable, indicating more suitable as reference genes.

geNorm analyses for the 18 selected genes are shown in Appendix A. Under cold stress, the expression of *CYP*, *UBC5B*, and *GAPC1* was the most stable in the leaf; the expression of *UBQ*, *CYP*, and *ACT7* fluctuated least in the root; and the expression of *HIS6*, *TUA4*, and *GAPC1* was the most stable in the stem. Meanwhile, in different okra tissues, the genes most consistently expressed were *HIS6*, *CYP*, and *TUA4*.

Under salt stress, the most stable expressed genes were *EF-1α*, *UBQ*, and *ACT7* in leaf; *UBC5B*, *EF-1α*, and *CYP* in the root; *TUA4*, *Eif*, and 60s in stem; and *HIS6*, *UBC5B*, and *60s* in different tissues. Under drought stress, *UBQ*, *ACT7*, and *GAPC1* showed the most stable expression in leaf; *UBQ*, *UBC5B*, and *ACT7* expressed most stably in the root; and *TUA4*, *Eif*, and *HIS6* expressed most stably in the stem. Meanwhile, the most stable expressed genes were *UBQ*, *Eif*, and *HIS6* in different tissues.

Under all treatments, *HIS6*, *CYP*, and *HIS1* in leaf; *ACT7*, *GAPC1*, and *HIS6* in the root; and *HIS1*, *UBC5B*, and *GAPC1* in the stem had the highest expression stability. Moreover, *HIS6*, *Eif*, and *UBQ* were the most stable reference genes expressed in okra.

V2/3 values of paired variables were all less than 0.15 under abiotic stress and different tissues, indicating that the appropriate number of reference gene combinations in okra was 2. In addition, the data showed that, across all samples, paired variations of V2/3, V3/4, and V4/5 were 0.123, 0.152, and 0.141, respectively, which indicated that V values were altered by the addition of the third and fourth gene, 3 or 4 genes could be selected according to the trend of V value (Figure 2). Therefore, for all samples, the optimal reference gene combination was *HIS6*, *Eif*, *UBQ*, and *ACT7*.

### 3.4. NormFinder Analysis of Reference Genes in Okra

NormFinder’s analysis showed that under cold stress, the most stably expressed genes were *GAPC1* (0.167), *CYP* (0.232), and *ACT7* (0.3) in the leaf (Figure 3A); *ACT7* (0.076), *HIS6* (0.202), and *CYP* (0.233) in the root (Figure 3B); and *GAPC1* (0.054), *Eif* (0.097), and *ACT7* (0.125) in the stem (Figure 3C). Meanwhile, *60s* (0.07), *HIS6* (0.107), and *EF-1α* (0.191) (Figure 3D) had lower stability values across different tissues. Under salt stress, *TUA4* (0.127), *Eif* (0.157), and *UBQ* (0.164) had the highest expression stability in the leaf (Figure 3E) and *UBQ* (0.117), *ACT7* (0.144), and *TUA4* (0.203) were the most stable genes expressed in the root (Figure 3F). The three most stable genes were *CYP* (0.128), *ACT7* (0.131), and *EF-1α* (0.14) in the stem (Figure 3G), and in different okra tissues, *Eif* (0.072), *GAPC1* (0.083), and *ACT7* (0.125) (Figure 3H) were the most stably expressed genes. Under drought stress, *ACT7* (0.095), *Eif* (0.106), and *HIS6* (0.114) had the highest stability of expression in leaf (Figure 3I); *HIS6* (0.056), *UBQ* (0.077), and *Eif* (0.102) were the three most stable genes in root (Figure 3J); *HIS6* (0.022), *GAPC1* (0.085), and *UBC5B* (0.122) were most stably expressed in the stem (Figure 3K); and NormFinder analysis identified that *UBQ* (0.059), *GAPC1* (0.062), and *HIS6* (0.062) (Figure 3L) were found to be the top-ranked gene with stable expression in different tissues.

Under different treatments, *TUA4* (0.059), *60s* (0.102), and *HIS1* (0.143) were the most stable genes in leaf (Figure 3M); *Eif* (0.104), *HIS6* (0.15), and *TUA4* (0.172) were found to be suitable reference genes in the root (Figure 3N); and *60s* (0.059), *Eif* (0.073), and *GAPC1* (0.091) were among the most stable genes in stem (Figure 3O). Across all samples, the most stable genes included *TUA4* (0.134), *60s* (0.147), and *GAPC1* (0.187) (Figure 3P).

### 3.5. BestKeeper Analysis of Reference Genes in Okra

BestKeeper analysis showed that under cold stress, the most stable genes included *HIS6* (0.97 ± 0.29) and *EF-1α* (1.02 ± 0.30) in the leaf, *GAPC1* (0.66 ± 0.11) and *CYP* (0.88 ± 0.11) in the root, and *CYP* (0.83 ± 0.07) and *HIS6* (1.09 ± 0.17) in the stem. In different okra tissues, *Eif* (0.59 ± 0.18) and *UBQ* (1.04 ± 0.33) were the most stable genes. Under salt stress, the expressions of *HIS6* (0.62 ± 0.03) and *ACT7* (0.91 ± 0.22) were the most durable in the leaf, and the expressions of *HIS6* (1.07 ± 0.08), *EF-1α* (1.15 ± 0.12) were the most durable in the root. *Eif* (0.59 ± 0.16) and *UBQ* (0.91 ± 0.27) were ranked as the two most stable reference genes in the stem, and in different okra tissues, *Eif* (1.25 ± 0.35) and *HIS6* (1.25 ± 0.36) were the most stable genes. Under drought stress, the most stable genes included *UBQ* (0.99 ± 0.28) and *GAPC1* (1.09 ± 0.29) in the leaf, *GAPC1* (0.75 ± 0.23) and *ACT7* (0.99 ± 0.28) in the root, *Eif* (0.81 ± 0.19) and *UBQ* (0.86 ± 0.23) in the stem, and the expressions of *UBQ* (1.37 ± 0.09) and *Eif* (1.45 ± 0.15) were the most stable in different tissues.

Under different treatments, *ACT7* (0.59 ± 0.17) and *HIS6* (1.14 ± 0.33) were the most stable genes in the leaf, *HIS6* (1.33 ± 0.37) and *60s* (1.52 ± 0.37) expressed stably in the root, and *UBC5B* (1.47 ± 0.47) and *Eif* (1.73 ± 0.51) were the most stable genes in the stem. Meanwhile, *HIS6* (0.85 ± 0.14) and *Eif* (0.87 ± 0.19) were found to be the most stable genes in different okra samples (Appendix A).

### 3.6. RefFinder Analysis of Reference Genes in Okra

RefFinder analysis showed that (Appendix A), under cold stress, the most stable expression genes included *GAPC1* in the leaf, *CYP* in the root, *HIS6* in the stem, and in different okra tissues. *HIS6* was the most stable reference gene. Under salt stress, the most stable genes included *EF-1α* in the leaf and root, *TUA4* in the stem, and *HIS6* expression was most persistent in different tissues. Under drought stress, *UBQ* in the leaf, *HIS6* in the stem, and *TUA4* in the root were calculated as the most stable genes, and in different okra tissues, the most stable gene was *UBQ*. Among all treatments, the most stable genes included *HIS6* in the leaf and root and *UBC5B* in the stem. Moreover, *HIS6* was stable in all samples, and the most unstable gene was *TUB-α.*

### 3.7. The Validation of Selected Reference Genes Based on AeMYB1R1

To verify the reliability of the above algorithm results, the top two and bottom two genes for expression stability were used to calculate the relative expression level of AeMYB1R1 transcription factor.

Under cold stress, compared with the most stable genes (*HIS6*, *60s*) and their combination, the relative expression of *AeMYB1R1* reached a peak (9.38, 8.23, and 8.80) at 12 h. However, compared with the most unstable reference genes (*ARFA1E*, *TUB-α*), the relative expression level of *AeMYB1R1* exhibited a different trend (Figure 4A). In addition, under salt (Figure 4B) and drought stress (Figure 4C) and in other tissues (Figure 4D), compared with the most stable genes and their combinations, *AeMYB1R1* expression levels showed similar trends in variation. Conversely, compared to the two most unstable reference genes, *AeMYB1R1* expression levels and variation trends were inconsistent.

## 4. Discussion

Reference gene expression profiles vary with species, varieties, tissues, and stresses [17,44,45]. The selection of suitable reference genes is the key to the study of target gene expression in various experimental conditions and tissues. Therefore, it is crucial to screen for the most suitable reference genes for RT-qPCR analysis, and this study aimed to find the optimal reference genes for okra under abiotic stress. Few studies have evaluated reference genes for okra [31]. At present, this is the first specific report to verify okra reference genes under different abiotic stresses. In this study, 18 potential reference genes were selected from our published okra transcriptomic database, *CYP* and *ACT7* were more suitable in the root under cold stress, *EF-1α* and *HIS6* were more ideal in the root under salt stress. At the same time, *TUA4* and *Eif* were more suitable in stem under salt stress (Appendix A). The results showed that there were other optimal reference genes under different experimental conditions.

It was also evident in our findings that 18 genes showed differential expression in the root, leaf, and stem. Different reference genes should be selected even in various tissues of the same species, as is verified in most studies [46]. In *Toona ciliata*, *PP2C59* and *UBC5B* expressed stably in leaf and *HIS1* and *ACT7* showed good stability in young stem [23]. In different tissues (root, leaf, and flower) of woodland strawberries, the most stable reference genes were *FveMT1* and *FveSAND*, while *FveCHC1* and *FveEF1a* were more suitable for fruit [47]. In apple, *ACT* and *GAPDH* in bud, *EF-1*, and *PP2A* in flower, and *GAPDH* and *CYP* in fruit were the reference genes stably expressed [48].

Under different abiotic stresses, reference genes showed showed differential expression characteristics [45]. Of the 12 selected reference genes in *Peucedanum praeruptorum*, *PTBP1* and *PP2A* were stable under drought stress, and *TIP41* was the gene with an unchanging expression under salt stress. At the same time, *SAND* and *UBC9* exhibited the most stable expression under cold stress [49]. In parsley, under cold and salt stress, *EF-1α* and *TUB* were the most stable genes, and *ACTIN* expression was regular under drought stress [50]. In *Polygonum cuspidatum*, *60s* expression was regular under cold stress, while under drought and salt stress, the study found *ACTIN* and *TUB* to be the most stable genes, respectively [30]. In this study, the most suitable reference genes under cold stress were *HIS6* and *60s*; *UBQ* and *Eif* were most suitable under drought stress. At the same time, the expressions of *HIS6* and *Eif* were stable under salt stress (Appendix A). Therefore, it is essential to determine the optimal reference genes under different experimental conditions by RT-qPCR data and algorithm analysis. Histone deacetylases are responsible for removing acetyl groups from histone lysines, thereby activating gene transcription and regulating plant physiology, development, and environmental response [51]. In the present study, *HIS6* was expressed under different conditions, which may be a suitable reference gene under various abiotic stresses (Appendix A). In contrast, *TUB-α* was a gene with lower expression stability, which has also been proven in *Chrysoperla nipponensis* [52]. However, as a commonly used reference gene, *TUB* expression was stable under stress in *Betula platyphylla* [53], *Lymantria dispar* [54], and animals [55]. This also indicated that a specific reference gene has various expression stability in various species under different stresses. In addition, due to reasons such as primer design, extension time, false negative results, etc., RT-qPCR products that were too large or too small negatively affected data reliability, which should be considered in future research.

In this experiment, geNorm (Appendix A), NormFinder (Figure 3), BestKeeper (Appendix A), and RefFinder (Appendix A) were used for gene expression analysis. The first three algorithms were used to evaluate the stability of gene expression. However, due to differences in these software algorithms, the ranking results were inconsistent. For example, *UBQ* ranked first in the stability of gene expression in drought treated okra root by geNorm but ranked second and third in NormFinder and BestKeeper, respectively. In strawberry, *DBP* ranked first in different tissue and fruit development stages by geNorm but third by NormFinder and BestKeeper [56]. In *P. cuspidatum*, *60SrRNA* ranked first under abiotic stress by geNorm, but third and second by NormFinder and BestKeeper [30]. geNorm ranking is based on the similarity of expression levels of each reference gene from different experimental samples [25]. NormFinder evaluates the expression stability of reference genes based on intra- and inter-group variation [26]. BestKeeper directly relies on the CT value pairwise correlation analysis to calculate the stability ranking of candidate genes [27]. To combine the results of the first three algorithms, an online ranking software RefFinder was used for the overall ranking, which calculated the average gene weight and obtained the final ranking result [28]. To date, RefFinder has been applied in *Lilium regale* [57], *Brassica juncea* [2], and *Solanum rostratum* [58] for reference gene selection. In our study, according to RefFinder’s ranking, *HIS6* was the best reference gene in okra (Appendix A).

*AeMYB1R1* transcription factor is used as a reference for the control variables in this study, and *AeMYB1R1* was used to verify the expression level of 18 candidate genes. MYB proteins play an important role in plant development, metabolism, and response to biotic and abiotic stresses [34]. *MYB15* is involved in low temperature regulation and freezing tolerance processes in plants [36]. *AtMYB2* is involved in ABA induction of the genes that control salt and dehydration [59]. *MYB1R1* is involved in gene regulation under drought stress in plants [39].

## 5. Conclusions

This study first reported the selection and validation of reference genes in different okra tissues under other treatments. Under cold stress, the most stable genes in okra were *GAPC1* and *CYP* (leaf), *CYP* and *ACT7* (root), *HIS6* and *GAPC1* (stem), and *HIS6* and *60s* (different tissues). Under salt stress, the most stable genes included *EF-1α* and *UBQ* (leaf), *EF-1α* and *HIS6* (root), *TUA4* and *Eif* (stem), and *HIS6* and *Eif* (different tissues). Under drought stress, *UBQ* and *Eif* (leaf), *HIS6* and *Eif* (root), *TUA4* and *HIS6* (stem), and *UBQ* and *Eif* (different tissues) were the most stable genes. Furthermore, according to RefFinder’s analysis, the most stable genes were *HIS6* and *ACT7* in the leaf, *HIS6* and *Eif* in the root, and *UBC5B* and *60s* in the stem. Meanwhile, the most stable genes expressed in all samples were *HIS6* and *Eif*. The expression analysis results of *AeMYB1R1* also confirmed the validity of the results. These results provide suitable reference genes studying the molecular response mechanism of okra under abiotic stress and lay a foundation for the analysis of okra functional genes.

## Figures and Tables

**Figure 1 genes-14-00603-f001:**
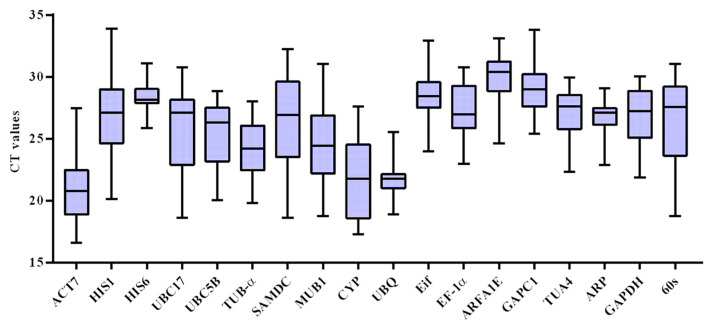
CT value distribution for 18 selected reference genes. The middle horizontal line represents the median.

**Figure 2 genes-14-00603-f002:**
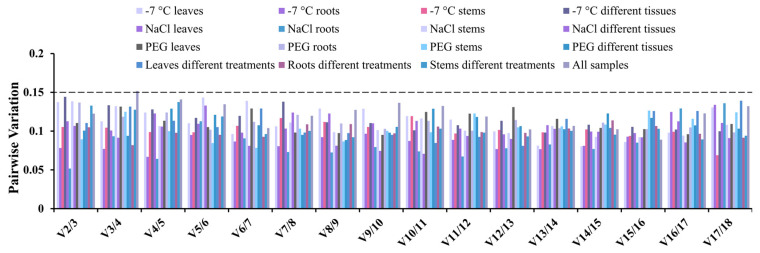
Pairwise variation value of 18 selected reference genes analyzed by geNorm. The threshold was set at 0.15, and the pairwise variation value greater than 0.15 indicated that more reference genes should be added for standardization.

**Figure 3 genes-14-00603-f003:**
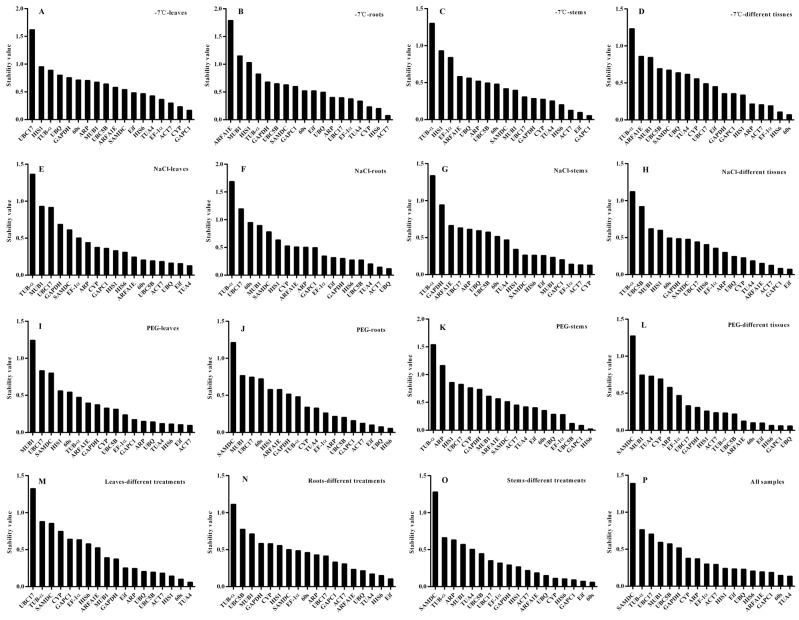
Expression stability ranking of 18 selected reference genes under different stresses and in different okra tissues, as analyzed by NormFinder. The stability value of the 18 reference genes in leaf (**A**), root (**B**), stem (**C**), different tissues (**D**) under cold stress, leaf (**E**), root (**F**), stem (**G**), different tissues (**H**) under salt stress, leaf (**I**), root (**J**); stem (**K**), different tissues(**L**) under drought stress, leaf (**M**), root (**N**), stem (**O**) under different treatments and all samples (**P**).

**Figure 4 genes-14-00603-f004:**
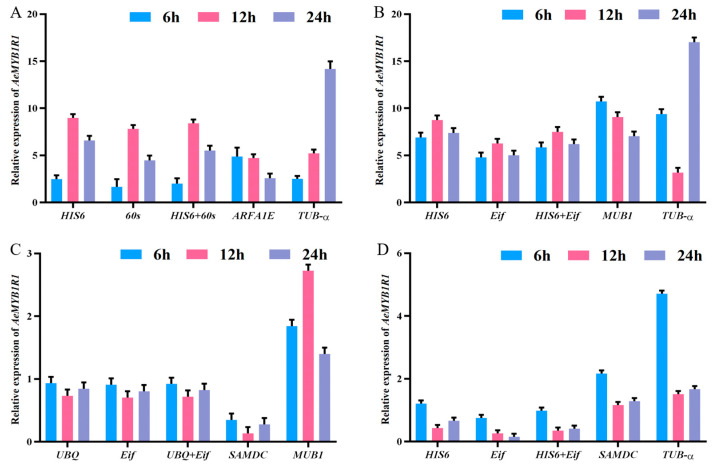
The relative expression level of *AeMYB1R1* determined by the most stable and unstable reference genes. Changes in *AeMYB1R1* transcription factor were normalized by selected reference genes (*HIS6*, *60s*, *HIS6 + 60s*, *ARFA1E*, *TUB-α*, *Eif*, *HIS6 + Eif*, *MUB1*, *UBQ*, *UBQ + Eif*, *SAMDC*) under cold (**A**), salt (**B**), and drought (**C**) stress, and in different tissues (**D**). Values are the means ± standard error of triplicate assays.

**Table 1 genes-14-00603-t001:** Experimental details of reference gene selection in okra.

Experimental Design	Treatment	Tissue	Biological Repetition	Sampling Points	Number of Samples
Cold stress	−7 °C	leaves, stems, roots	3	4	36
Salt stress	200 mM NaCl	leaves, stems, roots	3	4	36
Drought stress	20% PEG6000	leaves, stems, roots	3	4	36

**Table 2 genes-14-00603-t002:** The information of 18 selected reference genes and related primers, and PCR characteristics in okra.

Gene Symbol	Gene Name	Accession Number	Primer: Forward/Reverse	Amplification Product Size (bp)	Standard Curve	E (%)	R^2^
*ACT7*	Actin-7	OP448607	F:TCGCAGACCGTATGAGCAAG R:GGTGCTGAGTGATGCCAAGA	127	y = −3.5111x + 29.777	92.67%	0.9937
*HIS1*	Histone acetyltransferase MCC1	OP448608	F:GCTTCGGCACTTATCCACGA R:CCTCCGCACACACTTGAATG	138	y = −3.5239x + 30.469	92.21%	0.9974
*HIS6*	Histone deacetylase 6	OP448609	F:TGAGGCTTCTGGGTTTTGCT R:TTTGCCATTACCCACCCCAA	226	y = −3.4937x + 29.442	93.30%	0.9976
*UBC17*	Ubiquitin-conjugating enzyme E2–17 kDa	OP448610	F:TGCCGAGCTATACCCGATTG R:GGCCAACACCTTGCTTTCAC	190	y = −3.4411x + 26.954	95.26%	0.9971
*TUB-α*	Tubulin α-3 chain	OP448611	F:CCGAGAGCTGTGTTTGTGGA R:CGCCGACAGCATTAAACACC	241	y = −3.3538x + 29.262	98.69%	0.9967
*UBC5B*	Ubiquitin-conjugating enzyme E2 5B	OP448612	F:AGTGTTCCTTCCGCAACTTC R:CGCTTAGCTCTTCGTCGCT	188	y = −3.3125x + 25.576	100.39%	0.9929
*SAMDC*	S-adenosylmethionine decarboxylase proenzyme	OP448613	F:TGATGCCCTTGAGCCATGTT R:ACTTGAGTCTTGCCATCGGG	106	y = −3.3157x + 29.643	100.26%	0.9986
*MUB1*	Membrane-anchoredubiquitin-foldprotein 1	OP448614	F:CTTTCCGGCGGCTACAAGT R:ACATCACAAAGAGGGCTCTGG	173	y = −3.3915x + 28.576	97.18%	0.9979
*CYP*	cyclophilin	OP448615	F:CCGGAGGCGAATCCATCTAC R:AGACCCAACTTTCTCGACGG	224	y = −3.4893x + 30.282	93.46%	0.9968
*UBQ*	polyubiquitin	OP448616	F:CTGCACTTGGTCCTTCGTTTG R:GGAGCACCAAATGAAGAGTGG	244	y = −3.5635x + 29.459	90.82%	0.9998
*Eif*	eukaryotic translation initiation factor	OP448617	F:TGCTTGGCAATGGTCGATGT R:CCAGCAGCAATCCAAACCTTC	100	y = −3.5559x + 26.519	91.08%	0.9992
*EF-1α*	Elongation factor 1-α	OP448618	F:CCCGGTGACAATGTGGGATT R:GGCATAGCCGTTTCCGATCT	165	y = −3.4148x + 29.239	96.27%	0.9922
*ARFA1E*	ADP-ribosylation factor A1E	OP448619	F:CCCGGTGACAATGTGGGATT R:GGCATAGCCGTTTCCGATCT	165	y = −3.3776x + 25.795	97.73%	0.9997
*GAPC1*	Glyceraldehyde-3-phosphatedehydrogenase C subunit 1	OP448620	F:CGTGTCCCTACACCCAATGT R:TCGGTTGCTGTAACCCCATT	270	y = −3.4221x + 29.247	95.98%	0.9932
*TUA4*	Tubulin α-4 chain	OP448621	F:ATCTCAACCGCCTTGTCTCG R:CTCGGTACATGAGGCAGCAA	285	y = −3.2557x + 28.359	102.84%	0.9946
*ARP*	actin-related protein 2	OP448622	F:GGAGACGCATGTGCTGATTTG R:GTGACACCATCACCAGAGTCA	317	y = −3.3273x + 25.988	99.78%	0.9913
*GAPDH*	glyceraldehyde-3-phosphate dehydrogenase	OP448623	F:TGGAAGGATCGGTCGTTTGG R:CACCCCACGGAATTTCCTCA	239	y = −3.4732x + 29.391	94.05%	0.9964
*60s*	60s ribosomal protein l36	OP448624	F:CAACAAGGGCCACAAAACCA R:TTGACGATCTCGCGAACGAA	102	y = −3.5047x + 28.514	92.90%	0.9937
*MYB1R1*	Transcription factor MYB1R1	OM963000	F:CGCACCCATAACAACTCCCA R:TCTTTCACTTACTCCCTCTTCAGC	178	y = −3.4389x + 32.574	95.34%	0.9926

## Data Availability

The data presented in this study are available in Appendix A.

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
