# Peer review of "Selection and Validation of Reference Genes in Different Tissues of Okra (*Abelmoschus esculentus* L.) under Different Abiotic Stresses"

_genes, 2023, doi:10.3390/genes14030603_

Round 1
Reviewer 1 Report
This work describes the validation and selection of reference genes in okra under stress conditions. Several candidate genes were evaluated for their expression stability in different tissues of the plant under cold, drought and salt stress.
I would like to ask you if the RT-qPCR protocol is the same for all the primers used. In some cases (e.g. tubulin alpha-4 chain), the product of the amplicon is much more than 150 bp. In some cases, an extra time during the extension step may be necessary.
In addition, it is not clear if you used biological replicates for the RT-qPCR experimental procedure. Technical replicates are not enough to validate the stability of your results.
Author Response
This work describes the validation and selection of reference genes in okra under stress conditions. Several candidate genes were evaluated for their expression stability in different tissues of the plant under cold, drought and salt stress.
I would like to ask you if the RT-qPCR protocol is the same for all the primers used. In some cases (e.g. tubulin alpha-4 chain), the product of the amplicon is much more than 150 bp. In some cases, an extra time during the extension step may be necessary.
Response:
-- We thank the reviewer for this comment.
-- In this study, the qRT-PCR protocol is the same for all the primers used. As noted by the reviewer, the products of HIS6, TUB-α, CYP, UBQ, GAPC1, TUA4, ARP, GAPDH were more than 150 bp. After careful examination, we found that it was caused by the amplification characteristics of our designed primers. As suggested by the reviewer, some sentences have been added in paragraph 3 of the ‘Discussion’ section. Please review the modified sentences:
In addition, due to the reasons like primer design, the time during the extension step, false negative results etc., the RT-qPCR products that were too large or too small negatively influenced data reliability, which should be valued in future research.
In addition, it is not clear if you used biological replicates for the RT-qPCR experimental procedure. Technical replicates are not enough to validate the stability of your results.
Response:
-- We thank the reviewer for this comment.
-- Three biological replicates were used for the RT-qPCR experimental procedure in this study. As suggested by the reviewer, one sentence has been added in the ‘2.4 PCR and RT-qPCR analysis’ section. Please review the modified sentences:
Three biological and three technical replicates were run for each RT-qPCR run.

Reviewer 2 Report
The selection and validation of reference genes in different tissues of okra is a good idea. The authors utilized Okra tissue under different abiotic stresses to make this analysis. The idea is good, but the manuscript is overwritten. It is important to focus on main objective, develop a focused article and remove unnecessary additions.
I recommend to the authors to regroup and remove unnecessary and overwritten M&M and text in Results. For instance, Figure 2 can be just stated in the text and no need to show stability since there is not much difference seen in this figure. Figure 3 needs to be of a better quality.
Tables 3 and 4 can be made as supplementary material.
English usage throughout the text must be seen by an English language expert.
Author Response
The selection and validation of reference genes in different tissues of okra is a good idea. The authors utilized Okra tissue under different abiotic stresses to make this analysis. The idea is good, but the manuscript is overwritten. It is important to focus on main objective, develop a focused article and remove unnecessary additions.
I recommend to the authors to regroup and remove unnecessary and overwritten M&M and text in Results.
Response:
-- We thank the reviewer for this comment.
-- As suggested by the reviewer, overwritten words and sentences have been removed from the modified manuscript. Please review the modified paragraphs in ‘M&M’ and ‘Results’ sections.
For instance, Figure 2 can be just stated in the text and no need to show stability since there is not much difference seen in this figure. Figure 3 needs to be of a better quality.
Response:
-- We thank the reviewer for this comment.
-- As suggested by the reviewer, Figure 2 has been moved to the Supplementary Figures and renamed as Figure S2.
-- The DPI of Figure 3 has been improved from 150 to 300. At the same time, the structure and associated text have been modified to improve the quality of Figure 3.
Tables 3 and 4 can be made as supplementary material.
Response:
-- We thank the reviewer for this comment.
-- As suggested by the reviewer, Table 3 and 4 have been moved to the Supplementary Tables.
English usage throughout the text must be seen by an English language expert.
Response:
-- We thank the reviewer for this comment.
-- As suggested by the reviewer, the text has been thoroughly checked and the language has been polished by the editing service company. Please review the changes in words and sentences of the modified manuscript.
